# Shifting and Expanding Clause Combining Strategies in Heritage Turkish Varieties

Onur Özsoy [1,*] , Kateryna Iefremenko [2] and Christoph Schroeder [2]

1    Leibniz-Center General Linguistics (ZAS), 10117 Berlin, Germany
2    Institution of Germanic, University of Potsdam, 14469 Potsdam, Germany
*    Correspondence: oezsoy@leibniz-zas.de

**Abstract:** Turkish is a language described as relying predominantly on non-finite subordination in the domain of clause combining. However, there are also strategies of finite subordination, as well as means of syndetic and asyndetic paratactic clause combining, especially in the informal settings. Clause combining is and has been one of the focal points of research on heritage Turkish (h-Turkish). One point is particularly clear: In comparison with the monolingual setting, finite means of clause combining are more frequent in h-Turkish in Germany, the U.S., and the Netherlands, while non-finite means of clause combining are less frequent. Overall, our results confirm the findings of earlier studies: heritage speakers in Germany and the U.S. prefer paratactic means of clause combining using connectors, as opposed to monolingual speakers. Our results also reveal that age (adolescents vs. adults) and register (informal vs. formal) significantly modulate the use of connectors. Moreover, we find that the shift in preferences in means of clause combining triggers an expansion in the system of connectors and leads to the development of new narrative connectors, such as *o zaman* and *derken*. The system of syndetic paratactic clause combining is expanding in heritage Turkish. This expansion calls for multifaceted modeling of change in heritage languages, which integrates language-internal factors (register), dynamics of convergence with the contact languages, and extra-linguistic factors (age and language use).

**Keywords:** bilingualism; corpus linguistics; heritage languages; heritage Turkish; clause combining; coordination; connectors

## 1. Introduction

### 1.1. Heritage Languages

A heritage language (HL) is a language acquired at home in a society where it is not the majority language (ML) ([Lohndal et al. 2019]). Heritage speakers (HS) acquire the HL in their family, and they may acquire the ML as their (early) second language, or they acquire both languages simultaneously. Thus, usually at the beginning of the speakers' life, the HL is the dominant language of the HS, or the two languages develop simultaneously. However, with the speakers' gradual entry into the institutions of the society they live in, the ML becomes the dominant language especially in formal settings, since the ML is usually the only language they receive formal education in. Owing to these dynamics, language contact between the two languages is ubiquitous, in most cases, both in individual linguistic repertoires and in the speech community.

We investigate here heritage Turkish in Germany and in the U.S. In both countries, Turkish is an HL due to migration movement from Turkey (and partly also Cyprus). However, the sociolinguistic profiles of the Turkish-speaking communities in the two countries differ from each other. Migration from Turkey to Germany started around 60 years ago, and Turkish in Germany is described as a vital language, compared to other countries where it is an HL ([Yağmur 2011]). There are several reasons for that, and we name only those that we consider most important. First, the Turkish-speaking

community in Germany is large (around 4.3 million of Turkey-related population, according to Statistisches Bundesamt 2002) and lives closely together, especially in urban areas. Thus, Turkish-speaking people have an opportunity to use Turkish not only in their families, but also in informal public domains (cafes, shops, leisure-time activities etc.), at least in the larger cities. Second, Turkey is located relatively close to Germany, and people of Turkish origin have the opportunity to visit the country quite often, just as their relatives from Turkey may come to visit them in Germany. Third, Turkish as an HL is taught at some public schools (Küppers et al. 2015). Finally, Turkish media are easily accessible in Germany via multiple channels such as Turkish radio stations in Germany and satellite TV.

The sociolinguistic profile of the Turkish-speaking community in the U.S. is different in comparison with Germany. Not only is it smaller in numbers (it is estimated at 500,000, according to Grabowski 2005), but is also scattered throughout a much bigger country. Consequently, the Turkish language is mainly spoken only at home. Moreover, Turkish as an HL is not taught at public schools; Turkish HS in the U.S. only have an opportunity, in a few places (e.g., New York City), to study Turkish in Sunday schools organized by Turkish embassies and other authorities. Last but not least, the distance to Turkey plays a role, thus visits from and to Turkey cannot be frequent.

### 1.2. The Present Study

One of the central questions in heritage language research is whether the attested dynamics of variability, variation, and change can mostly be attributed to cross-linguistic influence, internal developments (Poplack and Levey 2009), or incomplete acquisition. In the case of cross-linguistic influence, the HL develops structures which resemble those of the dominant language of the society (transfer or convergence, cf. Montrul 2010). In the case of internal developments, we find a spread or further development of features of the informal spoken language, which is accelerated by the sociolinguistic environment, e.g., by a limited access to literacy, among other factors (Aalberse et al. 2019; Schroeder 2016; Wiese et al. 2022). In the case of incomplete acquisition, the dynamics are thought to be a result of low proficiency, limited exposure, and use (Montrul 2008; Polinsky 2008).

There are various preconditions for each of the three explanations to be considered in a given contact situation. That is, in order to consider cross-linguistic explanations, the structures of the dominant language(s) have to be in view. In addition, in order to discuss incomplete acquisition, the individual speakers' language proficiency, exposure, and use need to be taken into consideration. And in order to consider language-internal development, at least three factors need to be taken into account. One is register differences, which have to be controlled for (Wiese et al. 2022). Moreover, possible age differences need to be considered since these allow one to infer (or deny) generational change (Aalberse et al. 2019; following Labov and Harris 1994; also Tagliamonte 2016). Third, the sociolinguistic parameters of the speech community have to be taken into consideration. In a situation of higher vitality, we expect that novel forms are more readily available to the speakers (Labov 2012).

We address the research problem of language change in heritage Turkish with a focus on paratactic clause combining by means of coordinators in three varieties, i.e., heritage Turkish in Germany, heritage Turkish in the U.S., and monolingual Turkish in Turkey. Our more specific research questions are:

1. What are the differences between the three varieties in terms of frequency distributions and variation between strategies of clause combining?
2. What are the differences between the three varieties with respect to age, mode, and register variation in terms of the various clause combining strategies used?
3. Do we find in the heritage varieties shifting form–function relations in specific forms leading to the development of new coordinators? If yes, what are the reasons for their emergence?
4. To what extent can we relate our findings concerning the heritage varieties to internal developments, cross-linguistic influence, and/or incomplete acquisition?

In answering the first two questions, we follow a quantificational approach. In answering the third question, we follow a qualitative approach and investigate two particular forms in depth. The answer to question four follows correlational analyses which investigate the relationship between the use of novel connectors and extra-linguistic factors such as language use and proficiency.

The paper is organized as follows: In the remainder of this section, we develop the topic of clause combining, first from a typological perspective and then with a focus on canonical Turkish, concentrating on paratactic clause combining and the repertoire of connectors in Turkish. In Section 2, we describe our data. In Section 3, we present the results of our data analysis. In Section 4, we answer our research questions and discuss our findings with a view on previous research on heritage Turkish and in light of the contribution of previous findings to heritage language research in general.

### 1.3. Clause Combining

In his seminal typological paper on clause linkage, Lehmann (1988) develops three continua of clause linkage, namely (i) the continuum of integration of one clause into the other, (ii) the continuum of the expansion vs. reduction of the clauses, and (iii) the continuum of the mutual isolation vs. linkage of the two clauses.

We investigate primarily aspects of the first continuum here. One pole is that of parataxis, where two clauses are coordinated by means of some linking element and are syntactically independent from each other. The other is that of hypotaxis or subordination, where the two clauses are in a dependency relation to each other in the sense that one of the clauses has a syntactic function in the other. Languages may prefer finite embeddings here, as is the case with English and German, or they may prefer non-finite embeddings, as is the case with Turkish.

Between the two poles, we find a continuum of "hierarchical downgrading" (Lehmann 1988, p. 183f). Embeddings in core syntactic functions (i.e., complement clauses) are more tightly embedded than adverbial embeddings, and further adjoint clauses are more tightly attached than syntactically independent clauses which are coordinated by some coordinator.

Languages typically make use of a portfolio of clause combining strategies on this continuum (see, for example, the typological studies in Haspelmath 2004). Generally speaking, in a given communicative situation, the choice between a strategy which tends more towards the paratactic pole or one which tends more towards the hypotactic pole seems to depend very much on situational parameters. In ad hoc communicative situations of higher context-dependence, the frequency of paratactic linking structures increases, and in communicative situations of higher context-independence, the frequency of hypotactic linking structures increases. Since Chafe's early work on "information packaging" (Chafe 1976), this has been described in investigations of discourse structure (e.g., Miller and Weinert 2009) as well as in register-oriented approaches (Biber and Conrad 2009). Below we come back to the issue with an exclusive focus on Turkish.

### 1.4. Clause Combining in Turkish

Turkish is an agglutinating SOV language with a high variability of word order bound to information structural requirements. In sharp contrast to English and German, in canonical Turkish[1], subordination is realized mainly by means of non-finite forms. Complement clauses are clausal nominalizations; relative clauses are headed by participles; and adjunct clauses are headed by converbs or other non-finite verbs which are combined with postpositions. Non-finite subordination is preposed to its governing structure, i.e., the Turkish hypotaxis is "left-branching" (Johanson 1992, p. 267)—again in opposition to the contact languages English and German, where hypotaxis is right-branching, and subjunctors are positioned clause initially.

There are also a few means of finite subordination in Turkish, used mainly in informal registers. Further means of adjoint linkage include ellipses and conditional clauses, in

particular conditional clauses where the protasis is a wh-clause ('wh-conditionals') (Kornfilt 2014).

About the paratactic pole of the continuum, the phenomenon of interest for this paper, canonical Turkish certainly makes use of a variety of coordinators, i.e., lexical forms which serve to combine or coordinate syntactically independent clauses[2]. In particular, informal spoken registers of Turkey's Turkish seem to prefer paratactic means of clause combining (Kerslake 2007; Schroeder 2002, 2016), though the empirical evidence for this claim is still weak.

Turning to clause combining in heritage Turkish, this has repeatedly been the focus of attention in research. Generally speaking, hypotactic means of clause combining are said to be less frequent in heritage Turkish in Germany, France, and the Netherlands (Aarssen et al. 2001; Backus 2004; Treffers-Daller et al. 2006; Bayram 2013; Iefremenko and Schroeder 2019) in opposition to Turkish as used in Turkey. This generalization does not seem to hold for converbs, however (Treffers-Daller et al. 2006; Rehbein and Herkenrath 2015; Bohnacker and Karakoç 2020; Iefremenko et al. 2021). For heritage Turkish in the Netherlands, it has also been confirmed that finite means of subordination are preferred over non-finite means (Onar Valk 2015).

As for paratactic clause combining in heritage Turkish, Dollnick (2013) and Schroeder (2016) find a higher frequency of coordinators in the spoken and written language use of Turkish-speaking pupils in Germany as opposed to monolinguals in Turkey. Some studies also point to an emergence of the use of temporal-deictic expressions in the function of coordinators in heritage Turkish in Germany (Karakoç 2007; Herkenrath 2007; Herkenrath and Karakoç 2007; Herkenrath 2016). A particularly interesting form is the adverbial *o zaman* "then", identified by Herkenrath (2016) as an emergent connector in the speech of a group of Turkish–German bilingual children. Another form of interest here is *derken* "just then" which has previously been classified as a "discourse connector" (Göksel and Kerslake 2005, p. 453).

In identifying the class of connectors in canonical Turkish, we follow Johanson (2010; 2021, p. 768ff), who distinguishes between 'coordinative conjunctors' and 'adjunctors' or 'conjunctional adverbs'[3]. The first group is exclusively coordinators, while the second group is variable regarding their function as part of speech and may be used as adverbials and as coordinators. There is also a syntactic differentiation, i.e., adjunctors may combine with conjunctors, and in this combination, they must always follow (see also Şenlik 2022, p. 29 for Old Turkic).

The list in Table 1 gives all connectors we have found in our data, all varieties of Turkish alike. Note, however, that we exclude a possible connecting use of discourse markers here.

**Table 1.** Connectors in the Turkish data.

| Connectors | Adjunctors |
|:---:|:---:|
| *ama* "but" | *ancak* "even though" |
| *çünkü* "because" | *(bundan) dolayı(sıyla)* "consequently" |
| *de/da* "and" | *halbuki* "even though" |
| *eğer* "if" | *hatta* "even" |
| *fakat* "but" | *ondan* "because" |
| *hem . . . hem . . .* "both . . . and" | *üstelik* "furthermore" |
| *ne . . . ne . . .* "neither . . . nor" | *o zaman(da)* "then" |
| *ve* "and" | *o an(da)/o andan* "at that moment" |
| *veya* "or" | *(ama/daha/ve/ondan) sonra* "then" |
| *ya . . . ya . . .* "either . . . or" | *derken* "just then" |
| *yoksa* "or" | |

Some of the (coordinative) connectors coordinate other types of units as well as independent clauses, i.e., phrases or subordinate clauses. Some of the connectors (e.g., *ve* "and" and *hem . . . hem* "both . . . and") are always in clause-initial position; most, however,

are variable in their position and may be initial, integrated (following the topic of the clause, cf. Schroeder 2016, p. 93), or post-verbal (only in informal registers). In terms of frequency, however, the initial position is preferred over post-verbal and integrated (cf. Schroeder 2016 for *çünkü* "because").

It has to be noted that many of the coordinate connectors (including one subjunctor *ki*) are forms which have been borrowed from non-Turkic contact languages, mainly from Persian, but also Arabic (Johanson 1996; 2021, p. 768). Apparently, in earlier stages of Turkish, the system of converbal clause linking was more expansive than it is now (Wurm 1987). In addition, asyndetic clause combining was used in cases where in present-day Turkish a coordinator would be used (Johanson 1992, 1996, 2021). It is interesting to note that Wurm (1987) had already regarded present-day Turkish syntax as "Europeanized" in this respect.

## 2. Methods

### 2.1. Participants and Data

We analyze data from three countries, with two age groups in each, i.e., 33 adult and 32 adolescent HS of Turkish in Germany (Berlin), 27 adult and 34 adolescent HS of Turkish in the U.S. (New York City, New Jersey, and Washington), and 32 adult and 34 adolescent monolingual speakers of Turkish in Turkey (from the cities of İzmir and Eskişehir, both in the west). The age of the adult participants varied from 23 to 35 years, while adolescent participants were 15–18 years old (see Table 2). All HS were born and raised in Germany or the U.S. or had arrived there before the age of 5. Thus, most of the HS belonged to the 2nd generation, some even to the 3rd generation of immigrants. All of them speak Turkish (and sometimes the ML) at home and have no additional family language (e.g., Kurdish). Some of the HS participants received HL education to a limited extent; however, none of them attended a bilingual school. The adolescent participants were still in high school at the time of testing, while the educational background of adult participants varied from secondary school graduates to master's degree holders.

**Table 2.** Participant metadata.

| Country | Group | N | Age (Mean) | Gender |
|---|---|---|---|---|
| Turkey | adults | 32 | 27.63 | 11 females |
| | adolescents | 34 | 16.09 | 17 females |
| Germany | adults | 33 | 27.14 | 23 females |
| | adolescents | 32 | 16.00 | 17 females |
| USA | adults | 27 | 28.00 | 18 females |
| | adolescents | 34 | 16.00 | 22 females |

The data was elicited, transcribed, and compiled in a corpus within the frame of a larger research project "Emerging Grammars in Language Contact Situations: A Comparative Approach (RUEG)"[4]. The full corpus is in the process of being annotated for morphological, syntactic, and semantic features and is regularly updated for these layers (Wiese et al. 2021).

We use a snapshot of version 1.0 of the Turkish part of the corpus here. It includes 76,173 tokens in 768 documents. As each participant produced four sets of the same narration, this number of documents comes from 192 participants, roughly half of them adults and the other half adolescents (see Table 2).

### 2.2. Elicitation: Language Situations (Wiese 2020)

In order to elicit systematic data that are comparable across every country's cultural setting, participants took part in a complex semi-naturalistic narration task called the *Language Situations* method and described in Wiese (2020). The Language Situations method systematically elicits narrations allowing analyses of register and mode. Participants

first watch a video of a staged minor car accident that was filmed from a point-of-view perspective as if the viewer of the video was an actual eyewitness of the accident. After the participants watch the video, they are asked to retell the incident in four different communicative situations: to a friend via a WhatsApp voice message (informal spoken), to a friend via a WhatsApp text message (informal written), to the police via a voice mail (formal spoken), and to the police in the form of a written witness report (formal written). Each of these settings is accompanied by a change of rooms, i.e., a formally furnished room that looks like a government (police) office and elicitors who are dressed formally versus an informal looking room with cozy seating, cookies on the table, informally dressed elicitors, and informal chitchat about everyday issues and interests. The order in which these four different communicative situations were presented was randomly distributed across participants. At the end of the second session, participants were asked to fill out an online questionnaire jointly developed by the research unit RUEG[5]. The questionnaire comprises 10 sections: administration (with the information about the project number, country of elicitation, etc.), participants' general information, the educational background of the participants, the participants' languages, their family information, self-assessment of their language skills (in ML, HL as well as foreign languages, on a five-point scale), the participants' language use with family members and peers, a section concerning media use and free time (texting WhatsApp messages or writing emails in ML, HL as well as foreign languages, three scores of frequency), questions concerning personal character traits, and feedback on the participation in the study. There are two versions of the questionnaire, one for adults and one for adolescents. The latter one also includes questions regarding school and grade and excludes questions concerning jobs and spouses. The questionnaire was available in three languages; thus, participants from Germany filled it out in German, participants from the U.S. in English, and those from Turkey filled it out in Turkish.

Participants were recruited via personal networks and social media. The design of the study was explained to the participants before participation. They signed a written consent form which also clarified data protection issues according to the General Data Protection Regulation of the European Union. The study was approved by the ethics committee of the German Linguistics Society (Ethikkommission der Deutschen Gesellschaft für Sprachwissenschaft).

### 2.3. Statistical Analysis and Data Presentation

All our statistical analyses were run in R (R Core Team 2022). In addition to the base package of R, we used *tidyverse* for data manipulation and visualization (Wickham et al. 2019). We ran binomial generalized linear mixed-effects regression models using the *lme4* package (Bates et al. 2015).

We present our data in two ways. First, descriptive patterns and comparisons of groups will show the underlying tendencies in the data, e.g., that connectors are more frequently used by all HS groups. For a visual understanding of the data, we show several plots with direct comparisons of the U.S., Germany, and Turkey groups.

Second, we fit binomial generalized linear mixed-effects regression models to understand the effect of our independent variables (i.e., Country, Age, Register, Mode) on the binary outcome variable Connector (1) vs. Non-finite Verb (0) which is our dependent variable. Exchanging the predictors in our model, we probe if our predictors yield the best-fitting model to explain our data.

### 3. Results

We begin the results section with the overall pattern of clause combining in the data and then turn to the smaller and focused individual research questions. In other words, we first present how each participant group combines clauses using subordination versus coordination and then move on to a qualitative analysis of certain connectors which appear to be emerging forms in the heritage varieties of Turkish.

### 3.1. Overall Distribution of Non-Finite Subordination in Turkish Varieties

In Section 1, we introduced non-finite subordination as one of the main strategies available to combine clauses in Turkish. Below we present two plots that demonstrate the use of non-finite forms in our data: with the focus on the age of the speakers and with the focus on register and mode.

In Figure 1, we present the occurrence of non-finite verbs (nfV) in adults and adolescents in the three groups. The numbers are normalized per 100 tokens in the corpus. The red dots inside the box plots indicate the mean occurrence of non-finite verbs per group. We see that non-finite verbs are most frequent in Turkish monolinguals across both age groups, adults with mean = 9.00 nfV (SD = 4.53) and adolescents with mean = 8.43 nfV (SD = 4.05). The right-hand plot in Figure 1 shows that there is a cline in the adolescent group's nfV production which goes from Turkish monolinguals (nfV mean = 8.43; SD = 4.05) > heritage Turkish in the US (nfV mean = 6.64; SD = 3.48) > heritage Turkish in Germany (nfV mean = 5.35; SD = 2.88). However, the plot on the left-hand side shows that this is not generalizable across age groups since adult HS groups switch positions along the cline yielding Turkish monolinguals (nfV mean = 9.00; SD = 4.53) > heritage Turkish in Germany (nfV mean = 6.74; SD = 3.65) > heritage Turkish in the US (nfV mean = 6.31; SD = 3.06). Overall, the big stretch of the boxplots and the distribution of the per-speaker data points show that there is large variation in the use of non-finite verbs within each group which ranges from 0 occurrences to more than 22 per 100 tokens.

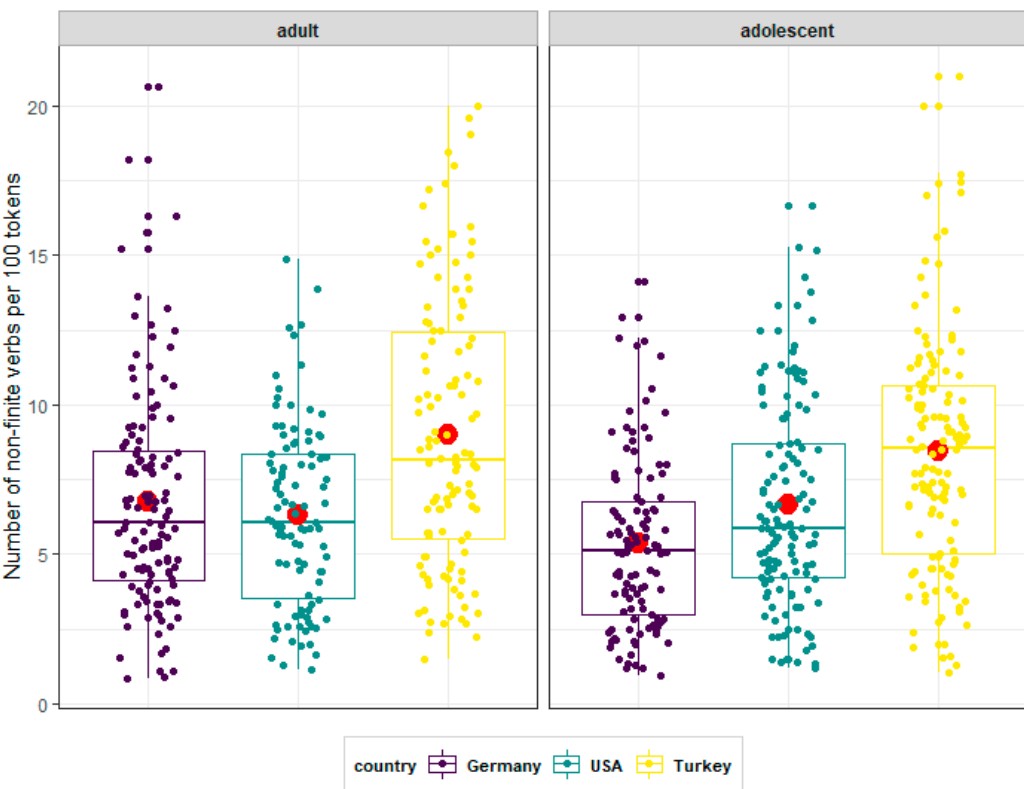

**Figure 1.** Box plots representing non-finite verbs per 100 tokens with the focus on age. Each data-point gives one speaker's use of non-finite verbs per 100 tokens.

Figure 2 presents the distribution of non-finite verbs by speakers from the three countries with the focus on register and mode. Visually, we can see that overall, HS use a smaller number of non-finite verbs in each communicative situation compared to the monolinguals. However, the plot also shows that in the monolinguals the frequency difference in the use of non-finite verbs between different communicative situations is apparent: the least number of non-finite verbs is used in the informal spoken communicative situation with mean = 6.04 nfV (SD = 2.81), followed by the informal written with mean = 6.98 nfV

(SD = 3.55), the formal spoken communicative situation with mean = 9.35 nfV (SD = 3.83), and the highest number of non-finite verbs is used in the formal written communicative situation with mean = 12.0 nfV (SD = 4.08). At the same time, even though the pattern in the frequency distribution in HS in Germany is similar to the monolinguals, the difference between the communicative situations is less obvious in this group: mean = 4.42 nfV (SD = 1.97) in the informal spoken, mean = 5.27 nfV (SD = 3.12) in the informal written, mean = 6.42 nfV (SD = 3.58) in the formal spoken, and mean = 7.98 nfV (SD = 3.43) in the formal written communicative situation. As for the HS in the U.S., Figure 2 shows that the difference between the communicative situations is almost absent: mean = 5.61 nfV (SD = 3.16) in the informal spoken, mean = 6.58 nfV (SD = 3.68) in the informal written, mean = 6.47 nfV (SD = 3.16) in the formal spoken, and mean = 7.32 nfV (SD = 3.07) in the formal written communicative situation. Thus, the differences in frequency distribution of non-finite groups between the HS groups and monolinguals seem to lie in the formal registers.

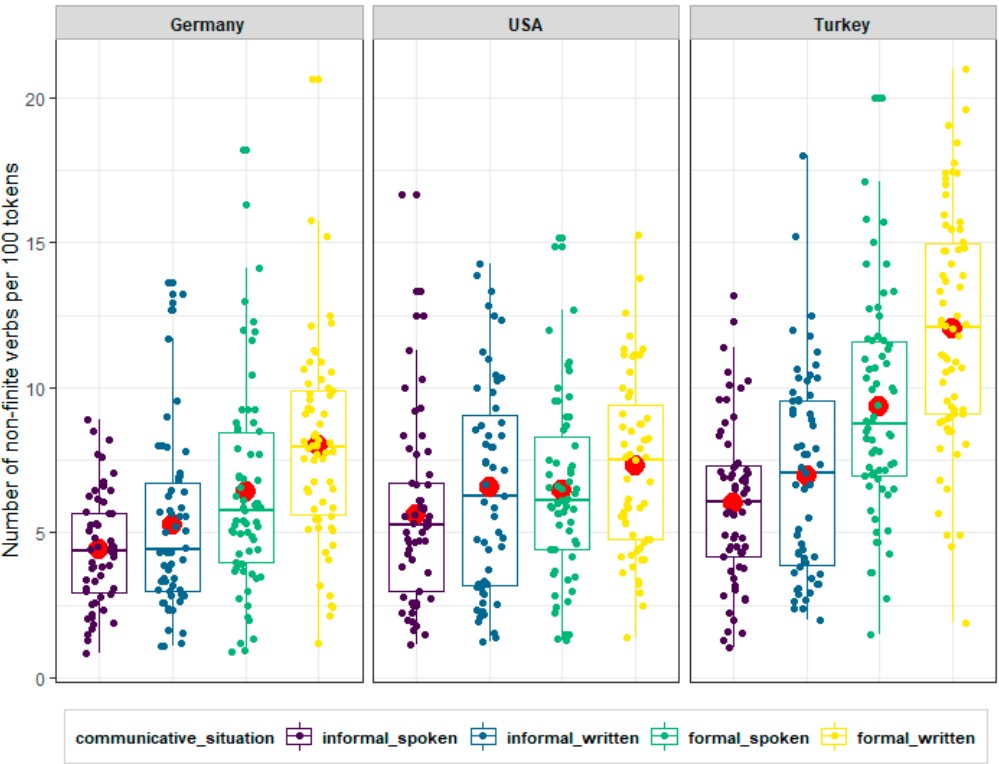

**Figure 2.** Box-plots that represent the distribution of non-finite verbs by speakers from the three countries with the focus on register and mode.

### 3.2. Overall Distribution of Coordination (Parataxis) Based on Connectors in the Turkish Varieties

Below we present the occurrence of clause combining connectors in our data with separate foci on age on the one hand and register and mode on the other. The numbers are normalized per 100 CUs in the corpus[6]. The red dots show the mean number of connectors used in each group. In the analysis, we included all connectors occurring in our corpus (see Table 1 and the discussion in Section 1.4 above).

In Figure 3, we see that connectors are more frequently used in Turkish HS as opposed to monolinguals across both age groups, adults and adolescents. The right-hand plot in Figure 3 shows that there is a cline in the adolescent group's connector production which goes from heritage Turkish in the US (connectors mean = 38.7; SD = 21.04) > heritage Turkish in Germany (connectors mean = 34.6; SD = 17.9) > Turkish monolinguals (connectors mean = 24.4; SD = 17.2). However, again the plot on the left-hand side shows that this is not generalizable across age groups since in the adult group the heritage groups switch

positions along the cline yielding heritage Turkish in Germany (connectors mean = 31.7; SD = 20.00) > heritage Turkish in the US (connectors mean = 28.2; SD = 15.5) > Turkish monolinguals (connectors mean = 21.6; SD = 16.5).

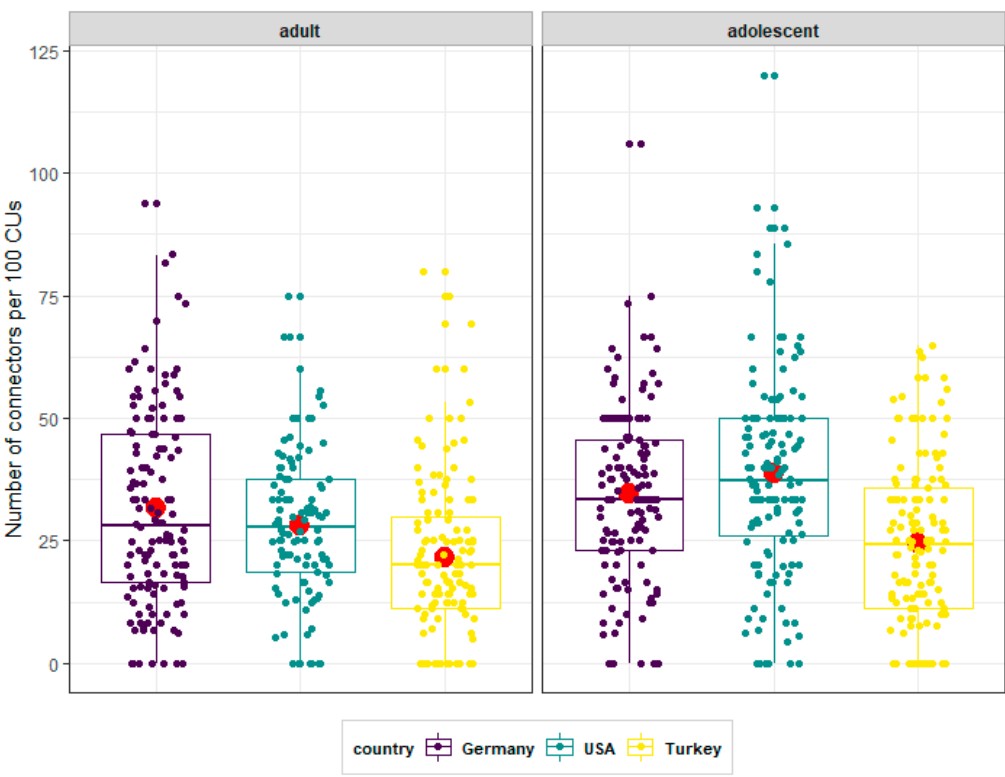

**Figure 3.** Box plots representing connector use per 100 CUs with the focus on age.

Figure 4 presents connector use per 100 CUs with the focus on register and mode. Again, we see that HS use connectors more frequently than monolinguals, but unlike the use of non-finite verbs, a higher use of connectors by the HS groups seem to hold for every communicative situation. Moreover, there is no apparent difference in the use of connectors across different communicative situations in the three country groups. For instance, monolinguals in Turkey use on average 19.9 connectors (SD = 15.3) in the informal spoken communicative situation, 19.0 connectors (SD = 16.5) in the informal written, 25.6 connectors (SD = 18.2) in the formal spoken, and 27.7 connectors (SD = 16.00) in the formal written communicative situation, while HS in Germany use on average 29.3 connectors (SD = 18.2) in the informal spoken, 28.3 connectors (SD = 17.2) in the informal written, 36.3 connectors (SD = 21.8) in the formal spoken, and 38.5 connectors (SD = 16.9) in the formal written communicative situation, and HS in the U.S. use on average 32.5 connectors (SD = 21.4) in the informal spoken, 35.6 connectors (SD = 20.4) in the informal written, 33.5 connectors (SD = 19.6) in the formal spoken, and 34.5 connectors (SD = 17.6) in the formal written communicative situation.

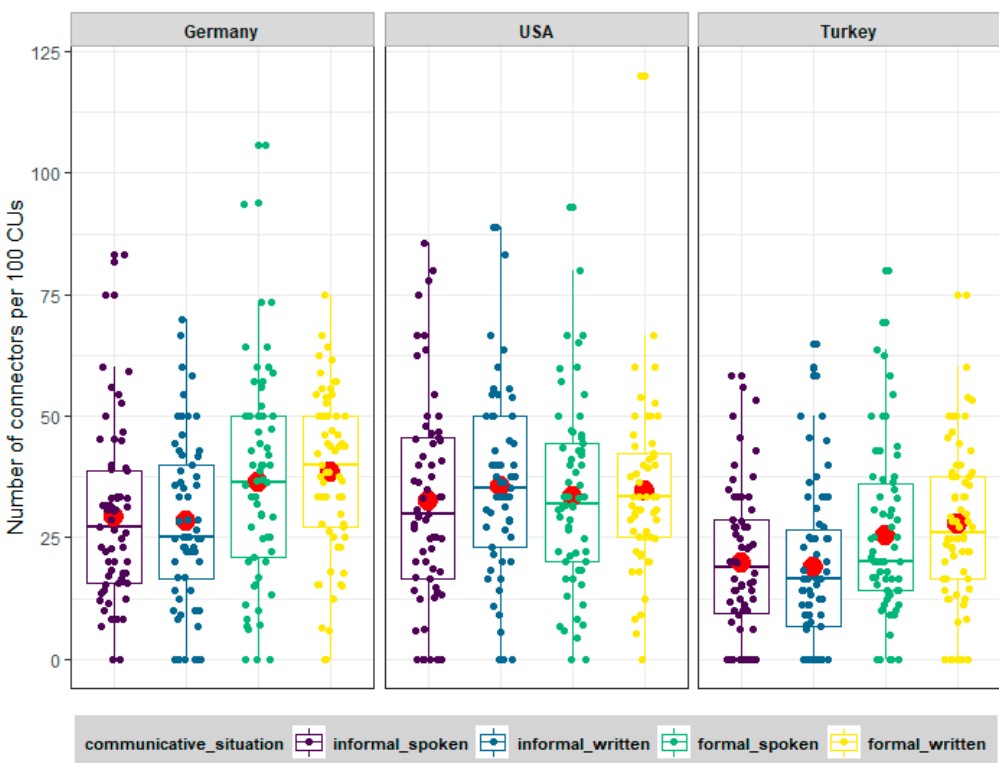

**Figure 4.** Box plots representing connectors use per 100 CUs with the focus on register and mode.

### 3.3. Modeling the Use of Connectors and Non-Finite Verbs

In a binomial generalized linear mixed effects regression model (GLMM), we tested the effects of four predictors (i.e., Country, Age, and the interaction between Mode and Register) on connector and non-finite verb use which is our binary-dependent variable. Here, we will describe the effect of each independent variable in this model. Table 3 gives an overview of the regression results.

**Table 3.** Regression table for binomial GLMM with the dependent variable Connector and the independent variables Country, Age, and the interaction of Register and Mode.

| Fixed Effect | β (σ) | *p*-Value |
|---|---|---|
| (intercept) | −0.7 (0.09) | *** |
| Country (Germany) | 0.35 (0.08) | *** |
| Country (Turkey) | −0.64 (0.08) | *** |
| Age (adult) | −0.28 (0.12) | * |
| Register (informal) | 0.27 (0.08) | *** |
| Mode (written) | −0.10 (0.07) | *ns* |
| Register*Mode | 0.19 (0.11) | *ns* |

$R^2$ = 0.18; significance levels: *** $p < 0.001$; * $p < 0.05$; *ns*: not significant.

**Country.** In our model, the baseline for this independent variable is the grand mean of all three groups since the Country levels are sum contrast coded. Our predictor Country yields significance on the population-level ($p < 0.001$). This implies that there are significant differences between the Turkish varieties in the three countries regarding the use of connectors compared to non-finite verbs. The direction of the effect depends on the Country level, as shown by a post hoc Tukey test which contrasts the three levels to each other separately. Table 4 shows the results of this test. In the first two rows, we see that the Turkey level shows a large significant negative effect (β = −0.99 and β = −0.92, respectively), meaning that speakers in Turkey tend to use significantly fewer connectors and more non-finite verbs than HS. In the test between the Germany and USA levels, we do not find a significant

effect[7]. Overall, we can note that the country levels Germany and USA facilitate the use of connectors, whereas the Turkey level does not.

**Table 4.** Post hoc Tukey test for the three levels of the Country variable.

| Country | β (σ) | *p*-Value |
|---|---|---|
| Turkey–Germany | −0.99 (0.14) | *** |
| Turkey–USA | −0.92 (0.14) | *** |
| Germany–USA | −0.07 (0.14) | *ns* |

Significance levels: *** $p < 0.001$; *ns*: not significant.

**Age.** Our study design sampled two age groups of speakers in each country, adolescents and adults. The exact numbers of participants in each group are reported in Table 2. In other words, we treat Age as a binary variable with the levels adolescent and adult. The Age variable significantly predicts that adults use fewer connectors with a moderate effect (estimate: −0.28, SE = 0.12, $p < 0.05$). In contrast, adolescents are more likely to use connectors.

**Mode.** Mode is a binary-independent variable with the levels spoken and written. Our model shows no reliable or significant effect of Mode (estimate: −0.10, SE = 0.07, $p > 0.05$).

**Register.** Register is a binary-independent variable assigned the levels formal and informal in our study. As seen in Table 3, the informal register significantly facilitates the use of connectors with a moderate effect (estimate: 0.27, SE = 0.08, $p < 0.005$). In contrast, the formal register limits the use of connectors.

**Interaction of register and mode.** As explained above, our study's design was separated into four communicative situations that involved participants' spoken and written narrations to a friend (informal) and spoken and written reports to the police (formal). For that reason, we tested the interaction of our predictors Mode and Register in our model. The interaction is not significant ($p > 0.05$).

In a nutshell, we can summarize the overall implication of the analysis of all the predictors like this: Our model shows that adolescent HS are most likely to use connectors especially in informal registers.

*3.4. Distribution of Specific Connectors in Turkish Varieties in General*

Our findings above suggest that HS of Turkish in Germany and the United States prefer coordination via paratactic clause combining compared to monolinguals in Turkey who showed a stronger preference for hypotaxis. An immediate consequence is the much higher frequency of connectors in the heritage varieties, as shown in Figure 4 above.

The narrative text type we elicited triggers a certain type of connectivity, namely that of temporal relations between events. Consequently, we are now in a position to shed light on the distribution of temporal connectors in the data. First, we present descriptive statistics in this subsection. Then, in the next subsection, we give qualitative analyses of novel functions of temporal connectors in heritage varieties.

The frequencies of these connectors are illustrated in the bar plots in Figure 5. From the color code used, we see that there is a clear preference for certain temporal connectors in all the varieties of Turkish, *sonra* (n = 223) "then" and its variant *ondan sonra* (n = 147) "after that" being the most frequent ones. Then we find *o anda* "at that moment" (n = 59), *o zaman* "then" (n = 32), and *derken* "just then" (n = 25) with relatively moderate frequencies in the corpus. Strikingly, the connector *daha sonra* "even later" is only present in the monolingual Turkish data. Further variants of *sonra* only show occasional presence in the data across all varieties of Turkish. Furthermore, there are two connectors which almost exclusively show up in the heritage varieties, namely *o zaman* "then" and *derken* "just then".

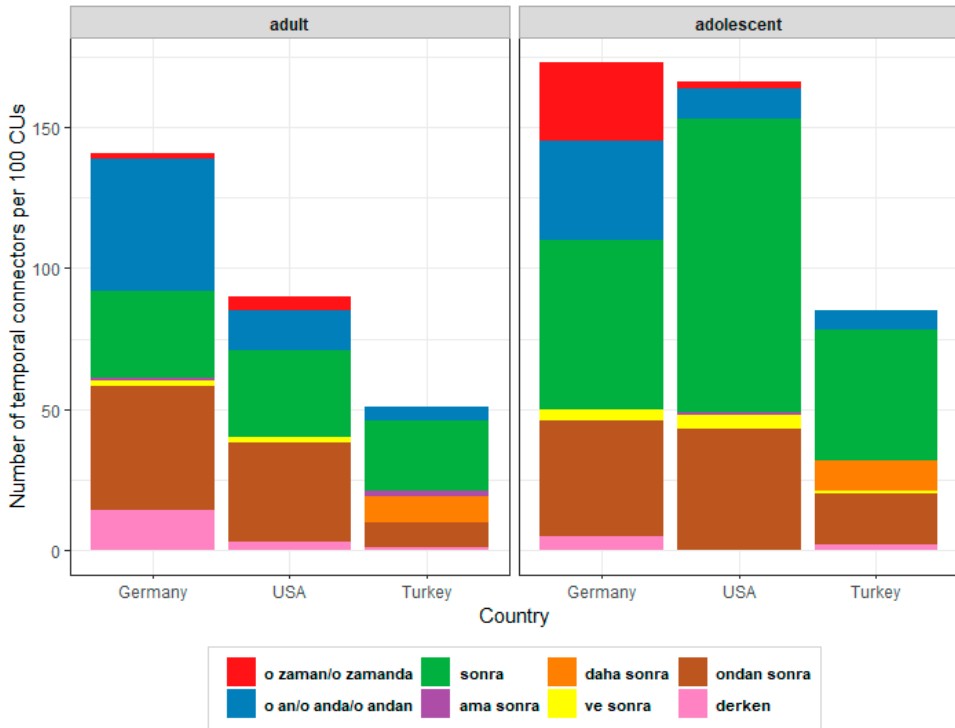

**Figure 5.** Bar plots representing the number of temporal connectors per 100 CUs.

Additionally, the box plot in Figure 6 affords insights into the distribution of temporal connector use. While the observation that HS use temporal connectors much more widely than monolinguals is generally valid, the height of the box plots and their interquartile ranges indicate that there is considerable variation among HS as compared to the monolinguals. However, all groups include a small number of outliers who produced up to 50 to 60 temporal connectors per 100 CUs.

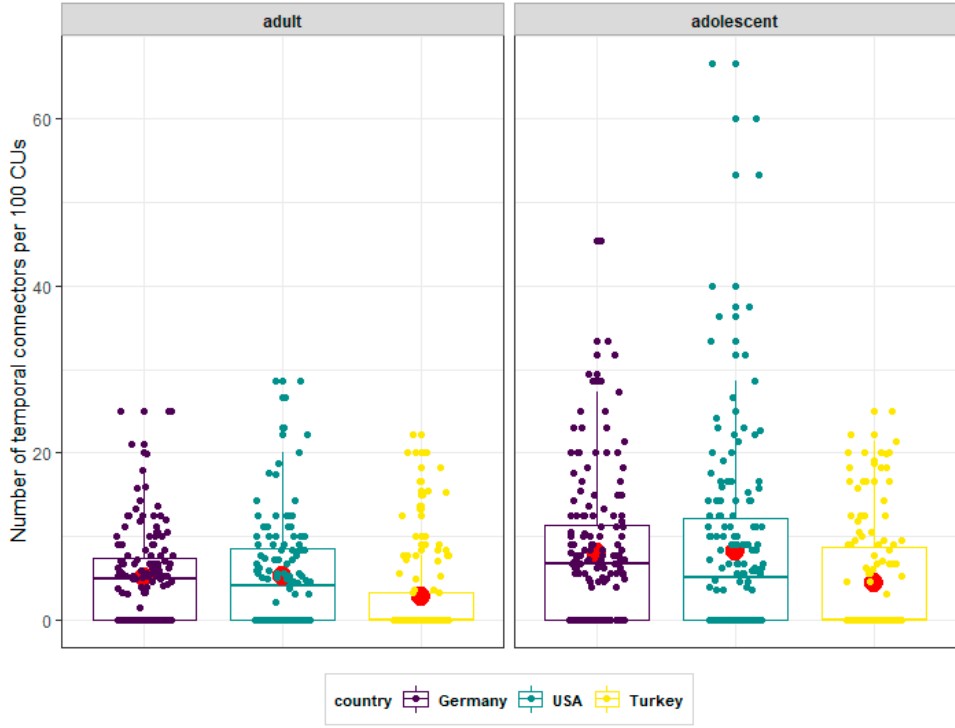

**Figure 6.** Box plots for individual participants' number of temporal connectors per 100 CUs.

### 3.5. Distribution of Functions of o Zaman and Derken

In order to understand possible changes in form–function relations of connecting elements in Turkish better, we now turn to a qualitative analysis of two connectors that are most characteristic in their distribution across different speaker groups: *o zaman* and *derken*. *O zaman* has previously been explored with a view on its connecting function in a qualitative study of heritage Turkish in Germany (Herkenrath 2016). However, we extend the scope of these observations given our controlled methods used to collect data and the direct elicitation of narratives that necessitate the use of connectors. *Derken* has not received any attention from the heritage language literature so far.

#### 3.5.1. o zaman

The form *o zaman* has been described to have two functions that are widely established in canonical Turkish (Göksel and Kerslake 2005). One is its function as an adverbial introducing a turn in a dialogue, where one speaker's beliefs are revised based on information retrieved from another speaker or another source of information. Consider the following dialogue in the given context:

Ahmet comes home from a session at the gym and sees that the bathroom lights are on. He presupposes that someone is using the bathroom and that this is the reason for the lights being on. Then, he meets his roommate Zeki in the kitchen and has the following dialogue:

(1)   A: Çok terledim. Z: Duş müsait. A: *O zaman* bir duş alayım. *"A: I'm very sweaty. Z: The shower is free. A: Then, I'll take a shower."*

Ahmet's beliefs of the occupation status of the bathroom were renewed based on the new information. He signals this by announcing the upcoming action with the consequence operator function of *o zaman*. Since our dataset consists only of monological narrations, no use of this function of *o zaman* is found.

Another widely established function of *o zaman* is again an adverbial use, where *o zaman* refers to a particular point in the past that has already been introduced in the narrative. This has a refocusing function as seen in example (2) from the corpus.

(2)   Adam topla oynuyordu ve top birden elinden kaydi ve yola düstü. Tam da **o zaman** bir araba geliyordu ve birden frenledi. (DEbi12FT_iwT)[8]

"The man was playing with the ball, and suddenly the ball slipped and fell on the street. Right **then**, a car was coming and suddenly braked."

In addition to these two clearly adverbial functions of *o zaman*, Herkenrath (2016) points to a new, connecting function which seems to be particularly present in heritage Turkish in Germany. Example (3) is representative of the novel form–function relation at hand. Here, *o zaman* serves to signal sequentiality in the narration. Hence, it is mainly a *narrative connector* rather than a temporal one.

(3)   Bir tane çocuk yolda yürürken o da e topla oynuyordu. **O zaman** top da e sokağa fırladı. **O zaman** yanında köpek vardı. (DEbi75FT_fsT)

"While a child was walking on the street, s/he was playing with a ball. **Then**, the ball fell onto the street. **Then**, there was a dog next to them."

In Table 5, we summarize the three functions of *o zaman* in the corpus and provide the raw numbers for the overall occurrences of *o zaman*.

**Table 5.** Types and functions of *o zaman* in Turkish and uses in the RUEG corpus.

| Connector | Type | Function | n in RUEG Corpus |
|---|---|---|---|
| o zaman "then" | Consequence | refocuses on state of affairs at level of action and discourse space, condition for further actions | 0 |
| | reference in past | refocuses on a point or extended stretch of time in self-experienced past | 10 |
| | narrative connector | refocuses on parts of previous narration announcing progression of narration | 23 |

Figure 7 shows that *o zaman* is only present in the heritage Turkish data but not in monolingual Turkish. It also shows that the narrative connector *o zaman* is more frequent in Germany as opposed to the U.S. Zooming even more into the use of *o zaman* as a narrative connector, the violin plot in this figure shows that while most speakers have only used one or two instances of *o zaman*, there are two speakers in Germany, who have produced 8 and 14 instances of *o zaman*, respectively. We look at possible extra-linguistic correlations for the individual variation in Section 3.5.3.

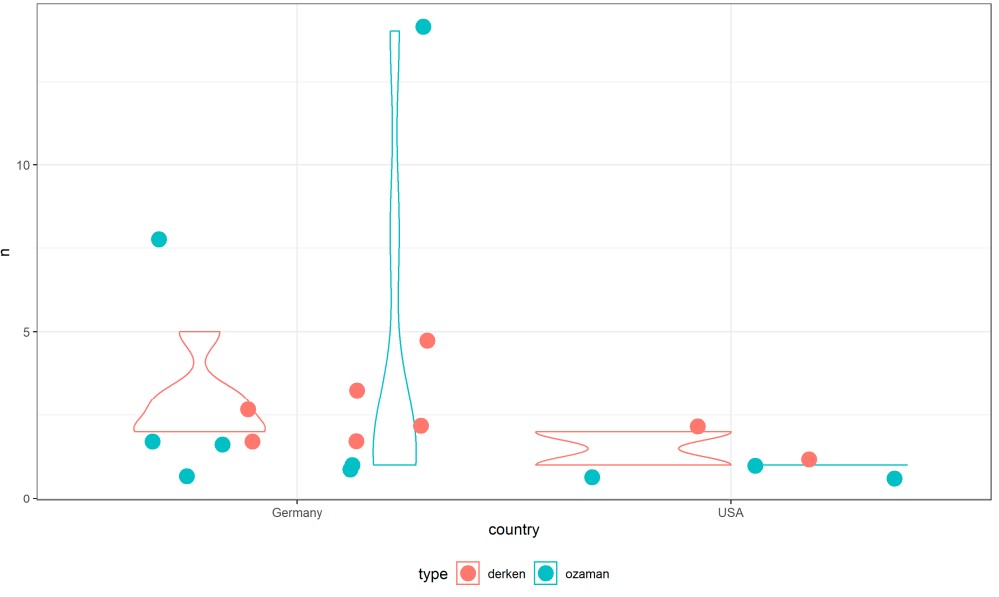

**Figure 7.** Violin plot for individual participants' use of the temporal connector *o zaman* and *derken*[9].

3.5.2. *derken*

Seeking further evidence for emerging form–function relations in heritage Turkish varieties, we look at *derken* "just then" which is referred to in passing as a "discourse connector" in Göksel and Kerslake (2005, p. 453)'s grammar of Turkish. It is a petrified (frozen) converbal form of the verbum dicendi *demek* "to say".

*Derken* seems to be used in three different functions. In the first use, it is not a connector but a postposition. As such, it establishes a phrase which appears clause initially and expresses the frame–ground/context for the following utterance ("speaking of x"). An instance of this use is example (4).

The second instance of *derken* is the prevalent use in Turkey's Turkish, and it is also the use which Göksel and Kerslake (ibid.) describe. Here, *derken* acts as a connector expressing a specific temporal-consecutive or final relationship between two events. It is important to note that just like in the first use of *derken*, it is part of the previous utterance, and it

preserves the quotative meaning which we also find in the first use. This is accompanied by the optative mode in the verb contained in the previous utterance. See example (5), where *derken* expresses a relationship where in the first utterance, an intention is conveyed, and the action expressed in the following utterance is a consequence of this intention.

In the third use, shown in example (6), *derken* is a clause-initial temporal connector. It expresses a relationship between two events where the first event evolves or continues, and the second event suddenly starts in that period.

(4)  yani çocuk da topla oynuyodu çocuk **derken** hani bi (-) yirmi yirmi (-) beş vardır (DEbi61FT_isT)

"So, the child was playing with the ball, **speaking of** child I actually meant someone around 20 or so."

(5)  köpek topa koşıyım **derken** sokağa fırladı (TUmo53MT_iwT)

"deciding to run for the ball, the dog threw itself on the street" (lit: "I will run for the ball **saying** the dog threw itself on the street"

(6)  bir genç çift tam yolu geçecekti (-) park alanında (-) **derken** ellerinden top düştü yola (DEbi04MT_isT)

"a young couple in the parking space was just about to cross the street **when** the ball fell from their hands on the road"

In Table 6, we summarize the three functions of *derken* in the corpus and provide the raw numbers for the overall occurrences of *derken*.

**Table 6.** Types and functions of *derken* in Turkish and uses in the RUEG corpus.

| Connector | Type | Function | n in RUEG Corpus |
|---|---|---|---|
| derken "just then" | frame-topic | establishes a previously introduced entity as the frame topic of the clause | 1 |
| | temporal-consecutive | the first event is about to evolve, and the second event suddenly affects it or is a sudden (intentional or unintentional) consequence of the first | 16 |
| | narrative connector | the first event evolves or continues, while the second event (suddenly) starts in that period | 8 |

In connection to Figure 7, we see again that the pure narrative connector function does not occur in our data from Turkey, but only in the heritage varieties, and again, it is most widely available and used in heritage Turkish in Germany.

### 3.5.3. Sources of Change in the Use of O Zaman and Derken: Incomplete Acquisition?

To understand if frequent and changing uses of *o zaman* and *derken* are possibly driven by factors which can be related to incomplete acquisition, we correlated the variable language use, language exposure, and self-rated proficiency from our participant questionnaire with the uses of *o zaman* and *derken* in the data. Language use refers to the frequency that participants spoke to people in their family and social circles in each of their languages. Language exposure measures how often participants heard Turkish by different people in their environment (e.g., parents, siblings, friends). Self-rated proficiency captures a score from 1 to 5 for each of the four core language skills (speaking, listening, writing, and reading, summing up to a maximum score of 20) in each of their languages.

None of these variables showed a meaningful correlation with the use of *o zaman* and *derken* (Appendix C). We exemplify this observation in Figure 8. The plot shows that

almost all of the speakers who used one of these two connectors reported a self-rating of a minimum 15 out of 20. This indicates that there is no meaningful relationship between the participants' Turkish proficiency levels and their use of these temporal connectors. On the contrary, almost all of them report to be highly proficient speakers of Turkish. Nevertheless, these speakers show newly emergent functions in the use of *o zaman* and *derken* which are unattested in analyses of monolingual Turkish in the literature and our data.

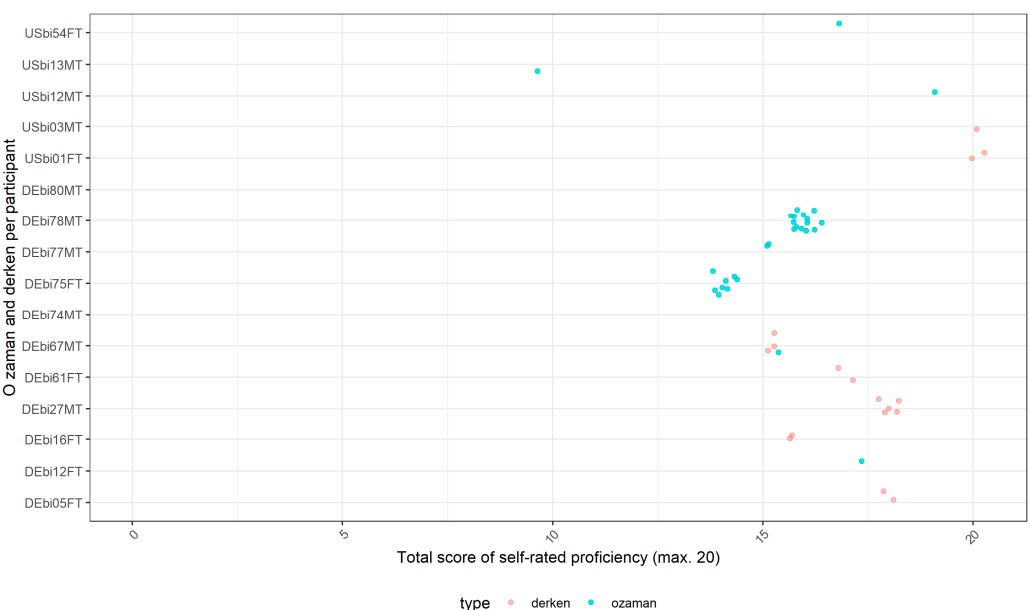

**Figure 8.** Jitter plot showing the total score of self-rated proficiency per participant regarding their uses of *o zaman* and *derken*.

## 4. Discussion

In the present paper, we investigated clause combining strategies in three varieties of Turkish, i.e., of HS in Germany and the U.S. and of monolingual speakers in Turkey.

The results of the frequency analysis showed that HS use a significantly lower number of non-finite verbs per 100 tokens compared to monolinguals. At the same time, the analysis showed that HS employ a significantly higher number of connectors compared to monolinguals. Such frequency distributions point to a gradual shift in the strategy of clause combining in heritage Turkish from non-finite subordination to paratactic clause combining accomplished by means of connectors.

All in all, our findings are in line with previous research on clause combining in heritage Turkish in Europe (Backus 2004; Treffers-Daller et al. 2006; Bayram 2013). However, we explore a new additional perspective here since we also consider age, mode, and register and show that preference for means of clause combining in canonical as well as heritage Turkish appears to be related to register. Clause combining by means of connectors is typical for informal settings, while non-finite subordination is preferred in formal settings (see Section 1). We showed that the difference between the groups is not significant in the informal settings. Only in the formal settings do HS use significantly lower frequencies of non-finite subordination compared to the monolinguals. Thus, we observe a spread of informal features of monolingual Turkish into the formal settings of heritage Turkish. Our findings then point to register levelling in HS in the domain of clause combining. HS acquire Turkish at home from their families and have very limited to no exposure to the formal setting of the language. Thus, it is not surprising that we find a spread of informal features in the formal settings.

Registers of a language are part of its internal repertoire, and the changes we observe here consequently support those approaches to language contact, which regard language-internal dynamics as a central dynamic in language contact (see Wiese et al. 2021; Tsehaye

et al. 2021; Schroeder et al. for HL, and Poplack and Levey 2009 for language contact in general).

Another significant predictor of the shift in preferences in the domain of clause combining is age. Our regression analysis showed that adolescent HS tend to employ connectors more frequently than adults. Moreover, looking closer at temporal connectors, we find that their usage frequencies amplify the overall trend for connector use—temporal connectors are particularly widely used by young HS. This finding is in line with sociolinguistic research that identifies adolescents as "forerunners of change" (Tagliamonte 2016). Adolescents are in a phase of life when they develop physically, socially, physiologically, intellectually, and linguistically. Their increasing independence, numerous new social contacts, separation from parents, and solidarity with peers impact the language they use.

Having presented frequencies of the means of hypotaxis and parataxis in our data, we zoomed in on temporal connectors because these are the type of connectors which are particularly frequent in our (narrative) data. We found that the general shift to more paratactic structuring seems to trigger an expansion of the function of specific connectors (*derken*) and temporal adverbs (*o zaman*) into more general clause-connecting devices without any temporal function (in the case of *o zaman*) or with a much more unspecific temporal function, which is also accompanied by a formal change (in the case of *derken*) from a (postposed) quotative form to a clause-initial narrative connector.

Overall, it seems that the higher functional load on paratactic clause combining in heritage Turkish in Germany and the U.S. leads to an expansion of the part of speech of connectors. In this respect, heritage Turkish syntax may be said to continue with a development already present in Turkey's Turkish and characterized there as "Europeanization" by Wurm (1987) (see the discussion in Section 1.4).

Furthermore, we have found that *o zaman* and *derken* in their novel function as connectors appear frequently in particular speakers, mostly from Germany. In order to check whether perhaps a low self-rated proficiency or low intensity of use of Turkish and exposure to Turkish may be the cause for the use of *o zaman* and *derken* in their novel function, we correlated our findings with the corresponding indices from the questionnaires. However, we have not found any correlative patterns in the data. This points to the fact that the use of such novel connectors is not the result of the speakers' 'incomplete acquisition', but are rather a product of a novel development in the heritage context, which is certainly triggered, amongst others, by limited access to the formal registers of the canonical variety. In this respect, our results provide counter-evidence to the 'incomplete acquisition' approach in line with Putnam and Sánchez (2013).

It is also important to emphasize that even though we did not find any significant differences in preferences for clause combining strategies between the heritage groups in Germany and the U.S., still, the emergence of novel connectors appears to be characteristic of heritage Turkish in Germany and is less acute in heritage Turkish in the United States. While transfer from the HS' majority German or English into their heritage Turkish seems to be a likely factor, it is not possible to identify a clear role for this. Both these MLs allow coordination with and without connectors just like Turkish does. In particular, the German translation equivalents of *o zaman* and *derken* are not outstanding in German in opposition to English, which makes it difficult to attribute these novel functions in heritage Turkish in Germany to language transfer per se.

Instead, we argue that the greater openness to change which we find in heritage Turkish in Germany may be linked to the higher vitality of heritage Turkish in Germany (or at least in Berlin, where we collected the data) compared to heritage Turkish in the United States. In contrast to heritage Turkish in the USA, heritage Turkish in Germany is also used in informal public social domains (see Section 1.1), and this leads to a higher inter-connectedness between speakers, which in turn promotes the spread of novel forms of language use. In line with Labov (2012), then, we interpret this higher vitality to facilitate the acceptance and use of novel connectors in a speech community.

More and more research on heritage Turkish varieties in different national and regional contexts promotes the idea that each variety should be treated and viewed in its own right. In addition to individual speaker differences, these group differences can lead to group level grammatical and lexical structures that are possibly unique for the respective variety. Thus, for multiple different heritage varieties that are in contact with Germanic languages (Bohnacker and Karakoç 2020 for Swedish; Onar Valk 2015 for Dutch; this study for English and German, among others), a continuum of varieties that generally prefer non-finite parataxis over finite hypotaxis can be defined.

Further corpus studies could investigate to what degree novel narrative connectors such as *o zaman* and *derken* are grammaticalized in heritage Turkish in Germany overall. Processing studies might show how and to what degree this novel function of *o zaman* is licensed. What remains certain is that as long as the vitality of Turkish in the given settings persists, changes in heritage Turkish varieties are likely to show up more frequently and prominently as contact-driven language change will be sped up with ongoing generations of heritage Turkish acquirers.

**Author Contributions:** Conceptualization, O.Ö., K.I. and C.S.; methodology O.Ö., K.I. and C.S.; formal analysis, O.Ö., K.I. and C.S.; data curation, O.Ö., K.I. and C.S.; writing—original draft preparation, O.Ö., K.I. and C.S.; writing—review and editing, K.I., C.S. and O.Ö; visualization, K.I. and O.Ö.; supervision, C.S.; funding acquisition, C.S. All authors have read and agreed to the published version of the manuscript.

**Funding:** This research was funded by the Deutsche Forschungsgemeinschaft grant 313607803 to GA 1424/10-1/ and SCHR 1261/4-1/. The APC were covered by the IOAP of the University of Potsdam.

**Institutional Review Board Statement:** The study was conducted according to the guidelines of the Declaration of Helsinki, and approved by the DGfS Ethics Committee of the Deutsche Gesellschaft für Sprachwissenschaft (German Society for Linguistics) (Corpus study: Protocol Code: #2017-06-171120, Date of approval: 20 November 2017).

**Informed Consent Statement:** Written informed consent was obtained from all participants involved in the study. In case of minors, their guardians provided their written informed consent to participate in this study.

**Data Availability Statement:** Our corpus data are available at https://zenodo.org/record/3236069. The data from the experimental study are accessible and publicly stored in an OSF repository which can be accessed on https://osf.io/f3hwc/ (accessed on 9 September 2022).

**Acknowledgments:** We'd like to thank Cem Keskin for his extensive feedback on an earlier version of this manuscript. We are grateful to two anonymous reviewers for careful reading and for pointing out some problematic points in previous versions. All responsibilities lie with us, of course. We also thank our research assistants Simge Sargın Kısacık, Gökçe Nur Taşagıl, Simge Türe, and Murat Uskan Oğuz who made this work possible with their detailed annotations of the corpus. We also acknowledge discussions with audiences at the 20th International Conference on Turkish Linguistics and the Turkic Linguistics Network Berlin-Brandenburg.

**Conflicts of Interest:** The authors declare no conflict of interest. The funders had no role in the design of the study; in the collection, analyses, or interpretation of data; in the writing of the manuscript, or in the decision to publish the results.

**Appendix A**

Export and coding of the Temporal Connector subset.

**For our** investigation, we need to know which temporal connectors are used in coordinated structures. Therefore, we defined a set of frequent Turkish temporal connectors as shown in Table 1. More specifically, paratactic clause combining by means of connectors typically appears clause initially in Turkish (Schroeder 2016). To make sure that our connectors were used in a conjunctive function, we limited the search to clause-initial uses.

To export the relevant data from the corpus, a search function in ANNIS (https://korpling.org/annis3c=rueg, accessed on 9 September 2022) looked like this: "_l_ cu

lemma=/o/ . norm=/anda/". We can paraphrase this as: Search for an object on the left of a communication unit (often a sentence). This object should form of the lemma "o" and the orthographically normalized word "anda".

Then, we exported relevant tiers such as *dipl* (=the orthographically or phonemically faithful transcription), *norm* (=the normalized transcription according to standard orthography rules), and *cu* (=the sentence layer including the sentence with the temporal connector and neighboring ones). These allow us to see the original spelling or pronunciation by the participant, the normalized spelling of each utterance by independent annotators (Turkish-speaking student and research assistants in the RUEG project), and the separation into communication units which allows us to see if a connector effectively appears clause initially in a coordinating function. More precisely, the GridExporter function with 20 words context to the left and the right was used.

All the exported temporal connector files were structured in a .csv with columns that include participant information (e.g., country, participant ID) and narrative information (i.e., register and mode of the narration). This file is accessible via the supplementary materials in the via the OSF repository of the project: https://osf.io/f3hwc/ (accessed on 9 September 2022).

*Statistical Analysis*

We ran a mixed effect linear regression model. The dependent variable is the normalized use of connectors as a continuous value. The independent variables in our model are Country, Age, Mode, and Register. Country encodes the assumed variety of Turkish, i.e., in which country the data were elicited. It has three levels which were sum-contrast coded assuming that all varieties of Turkish in our data are representative of the language; therefore each variable can be compared to the grand mean. Age is a binary variable with the levels adult and adolescent. Mode is another binary variable with the levels spoken and written. Situation is a binary variable that encodes informal and formal situations of elicitation in our data. We also included random slopes and intercepts for Participant.

**Appendix B**

**Table A1.** Germany. Regression table for binomial GLMM with the dependent variable Connector and the independent variables Age, and the interaction of Register and Mode. Outputs for by-Country binomial GLMM.

| Fixed Effect | β (σ) | *p*-Value |
|---|---|---|
| (intercept) | −0.37 (0.10) | *** |
| Age | 0.26 (0.10) | * |
| Register | −0.12 (0.05) | ** |
| Mode | 0.02 (0.05) | *ns* |
| Register*Mode | 0.05 (0.05) | *ns* |

$R^2 = 0.18$; significance levels: *** $p < 0.001$; ** $p < 0.01$; * $p < 0.05$; *ns*: not significant.

**Table A2.** Turkey. Regression table for binomial GLMM with the dependent variable Connector and the independent variables Age, and the interaction of Register and Mode. Outputs for by-Country binomial GLMM.

| Fixed Effect | β (σ) | *p*-Value |
|---|---|---|
| (intercept) | −1.31 (0.09) | *** |
| Age | −0.01 (0.01) | *ns* |
| Register | −0.30 (0.05) | *** |
| Mode | 0.08 (0.05) | *ns* |
| Register*Mode | 0.07 (0.05) | *ns* |

$R^2 = 0.18$; significance levels: *** $p < 0.001$; *ns*: not significant.

**Table A3.** USA. Regression table for binomial GLMM with the dependent variable Connector and the independent variables Age, and the interaction of Register and Mode. Outputs for by-Country binomial GLMM.

| Fixed Effect | β (σ) | *p*-Value |
|---|---|---|
| (intercept) | −0.43 (0.11) | *** |
| Age | 0.05 (0.11) | *ns* |
| Register | −0.16 (0.05) | ** |
| Mode | −0.07 (0.05) | *ns* |
| Register*Mode | 0.01 (0.05) | *ns* |

$R^2$ = 0.18; significance levels: *** $p < 0.001$; ** $p < 0.01$; *ns*: not significant.

**Appendix C**

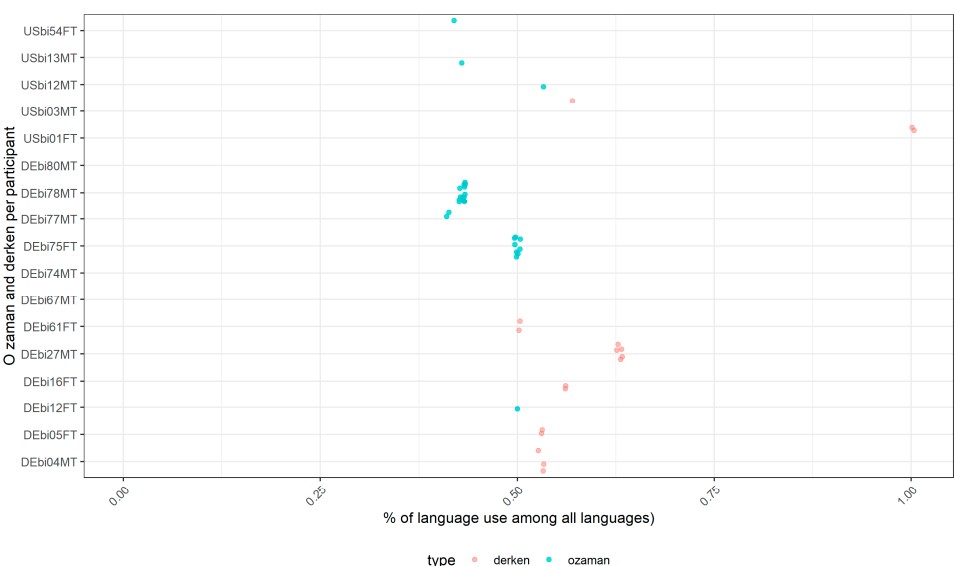

**Figure A1.** Jitter plot showing the total % of Turkish language use per participant regarding their uses of *o zaman* and *derken.*

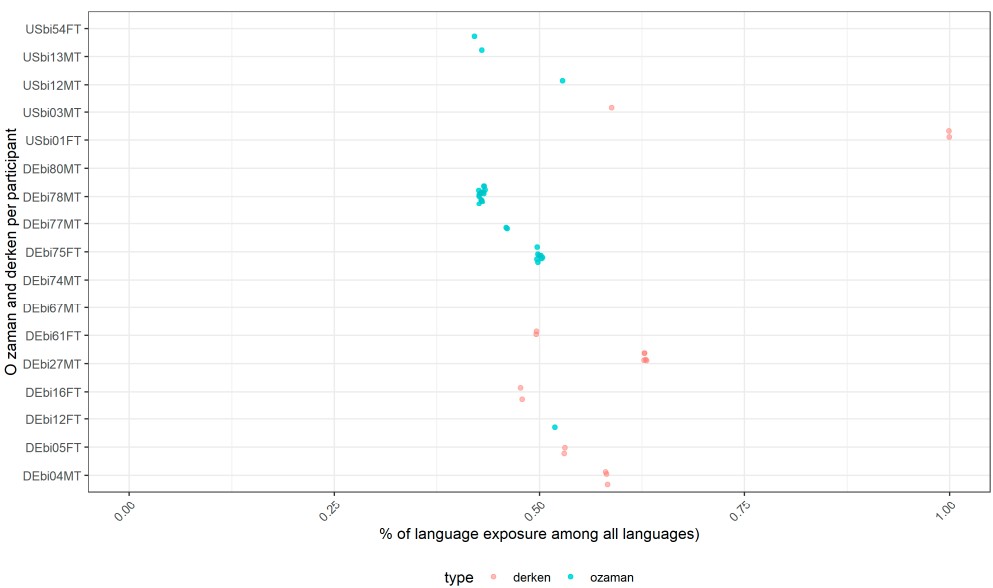

**Figure A2.** Jitter plot showing the total % of Turkish language exposure per participant regarding their uses of *o zaman* and *derken.*

## Notes

[1] With 'canonical Turkish', we refer to the standard varieties, spoken and written, as laid out in descriptive grammars such as (Kornfilt 1997; Göksel and Kerslake 2005; van Schaaik 2020).

[2] We take as a starting point Haspelmath's definition of coordination: "The term coordination refers to syntactic constructions in which two or more units of the same type are combined into a larger unit and still have the same semantic relations with other surrounding elements" (Haspelmath 2004, p. 29).

[3] Moreover, other grammars of Turkish follow this distinction, though with a different terminology, i.e., Göksel and Kerslake (2005, p. 212ff.: "conjunctions" vs. "discourse connectives"), (van Schaaik 2020, p. 339: "particles which carry a minimal amount of meaning" vs. "particles which connect two clauses in terms of reason, purpose, cause, and the like").

[4] Within this project, a narration task elicited data from more than 700 participants in five languages, i.e., German, English, Greek, Russian, and Turkish. Additional elicitation was conducted with other language pairs (e.g., Kurdish–Turkish–German) and other HL varieties (e.g., German in the United States).

[5] The PDF versions of the questionnaire can be found here: https://osf.io/qhupg/ (accessed on 9 September 2022).

[6] We normalized the use of connectors per 100 CUs because we assume that only one connector can be used per CU. However, as we see in Figure 3, there are several cases where there is more than one connector used per 1 CU (as demonstrated in the example below). The reason is that we calculated the overall number of connectors (clause-initial and not).

　　ve　　　　　sonra　　　　arkadaki　　araba　　gör-me-di　　　　　diye　　çarp-tı (USbi51MT_fwT)
　　and.CON　later.CON　behind　　car　　see-NEG-PST.3SG　CON　crash-PST.3SG
　　　　"In addition, then because the car in the back crashed because it didn't see"

[7] The post hoc result compares the levels of the Country variable separately to each other. In contrast, the full regression model with all the predictors importantly finds a significant effect of country. This is because the Country variable is sum-contrast coded meaning that each level is compared to the grand mean of all levels.

[8] The participant codes provide the following information:

- 　　Country: DE—Germany; TU—Turkey; US—U.S.;
- 　　Bi-/monolingual speaker: bi vs. mo;
- 　　Speaker number incl. age group: 1 to 50—adults; from 51 onwards—adolescents;
- 　　Gender: M vs. F (there were no speakers who identified as non-binary);
- 　　Heritage language for bilingual speakers or only family language for monolinguals: T for Turkish;
- 　　Communicative situation: formal—f vs. informal—i; spoken—s vs. written—w;
- 　　Language of production: T for Turkish.

[9] The plot only shows the uses of the novel functions of *o zaman* and *derken* in the heritage varieties. As there are no novel uses in the Turkey's Turkish variety and the other uses of *derken* are also extremely infrequent or did not occur in this group, the group is left out of the figure.

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
