# Peer review of "Shifting and Expanding Clause Combining Strategies in Heritage Turkish Varieties"

_languages, doi:10.3390/languages7030242_

Round 1

Reviewer 1 Report

This is a valuable paper which contributes original data and finding on varieties of Turkish used by speakers in the Turkish diaspora in the USA and Germany.  It is, for the most part clearly presented and appropriately placed in the context of previous research.  Nonetheless, a few, mostly minor points could be improved or discussed further.

Definition of "heritage language" as acquired at home..., line 51, should take into account previous wider usage e.g, Fishman 1999 "a language of personal relevance other than English", Fishman 2006: "languages other than the nationally dominant one that are historically associated with the ethnicity (the ethnocultural heritage) of particular minority populations".

The term "meta-linguistic" to refer to age and language use, line 25, might more appropriately be termed extra-linguistic as it doesn't pertain to the speakers' internal representations of linguistic aspects of Turkish.

The term "incomplete acquisition", line 74, is controversial and could be discussed, e.g, with respect to change in progress, their notions of attrition of social contexts for use, discussed in Putnam & Sanchez 2013. "what's so incomplete about incomplete acquisition".

Clarify the designation "canonical Turkish". Is that the Turkish described in grammars such as Kornfilt 1997 or Göksel and Kerslake 2005. It would be useful to refer to treatments of Turkish that deal with variation within varieties of Turkish in Turkey.  Kerslake (cited in text as 2007, line 172,  but in bibliography as 2004 line 801, discusses variation in informal spoken registers of Turkish in Turkey. 

Some other work on diaspora Turkish could be mentioned, e.g., Boeschoten 1990 ...,  Treffers-Daller & Sumru Özsoy 1995 ..., Aarsen 1996, Backus & Boeschoten (1998). Language change in immigrant Turkish. In G. Extra, & J. Maartens (Eds.), Mutilingualism in a multicultural context. Case studies on South Africa and Western Europe (pp. 221-237)   Aarsen/Akıncı/YaÄŸmur 2001 (clause linkage in NL, FR & Australia) ...

"Some of the HS participants received HL education to a limited extent" line 235. It would be useful to show this evidence in a table?  Turkish is taught as some public (and some private) schools in Germany but the proportion of children and adolescents with Turkish as a family language who attend these is quite low.   There is, however, much use of Turkish in the Turkish communities in Germany in informal (and semi-formal) contexts, as they note related to the population distributions of Turkish speakers in Germany vs. in the USA. (Anonymous et al. 2017).

Methods:

It would be helpful to have a footnote about the "risks and benefits" of the study, lines 267-268 which were presented to the participants along with the design of the study.

clarify what is meant by "pseudo-randomly" line 266.

The participant questionnaire for self-assessed proficiency, language use and language exposure is first mentioned in section 3, line 567. It would be appropriate to describe this questionnaire in section 2 on methods, and, perhaps, to include the relevant questions in an appendix.  Some differentiated studies of language use exist (Anonymous (et al.) 2008, 2015, 2018...

Results:

It seems that the two supposed binary predictors (factors?) register and mode turn out not to be independent but rather are discussed and presented together as communicative situation, with four values: informal_spoken, informal_written, _formal spoken and formal_written,  e.g., in Figure 1, line 341. This could be discussed more explicitly.

Speaker IDs could be explained explicitly in the text or in a footnote, IDs are missing from example (2) line 501 and example (6) line 555.

 Variety / Variation. The authors speak of "variety" in several places but there is considerable variation within the US and the DE populations, as shown in their figures; "varieties" would be more appropriate. Additional social or extra-linguistic information on the outliers would be interesting, if available, perhaps in a footnote or to be followed up in future publications. e.g. line 458.  For instance, on the two speakers in Germany who produced much higher frequencies of o zaman than the other speakers, line 519.

Since o zaman is far more frequent than derken, the authors might consider splitting the presentation of the results and discussion of these two connectors.

Section 4 Discussion

discussion of change over time, grammaticalization, processing studies -- could be elaborated. 

"What remains certain is that changes in heritage Turkish varieties are likely to show up more frequently and prominently as contact-driven language change will be sped up with ongoing generations of heritage Turkish acquirers"  lines. 688-690.  This is not so certain if, for instance, the use of 'Turkish declines over time so that primarily fixed phrases remain in the speech of "semi-speakers", c.f. the work of Dorian.

Terminology, especially statistical, could be explained more, perhaps with a list of terms and abbreviations in an appendix.  For instance "intercept" in Table 3, line 392.

"tighter embedded", "tighter attached" lines 139, 140 could be replaced by "more tightly",

"visualizes" is used in connection with the discussion of results in Figures, lines 310, 320, 339, 357 where "shows" or "presents" would be better.

footnote 7, below line 529,  "extremely limited to nonexistent" could be replaced by "extremely infrequent or did not occur"

"is not existent" line 560 could be replaced by "does not exit" or "does not occur".

Reviewer 2 Report

I think this is a very good paper and my recommendation is to either Accept it straight away or ask for Minor Revisions only. I have nothing critical to say really about the design of the study and the report on the results: it's all very clear and it all makes sense, and as a whole it contributes very useful new data to the narrow field of Heritage Turkish and the larger field of contact linguistics. My only comments are about this latter point: how the study fits in with the larger field of contact linguistics. If authors and editors feel it is useful, something more can be said about the implications of the study. This can all be done in the Discussion section.

The authors interpret the finding that Heritage Speakers tend to do clause combining in the same way in formal and informal registers as a case of register levelling, and that this is therefore a language-internal process. However, presumably this is not the entire story, since the levelling only happens in the bilingual settings. Presenting the levelling as language-internal suggests that contact has nothing to do with it, and this seems unwarranted: English and German have finite subordination and the contact situation involves reduced opportunities for getting enough exposure to and practice in the formal registers of Turkish. The same reluctance to attribute the variation at least partly to language contact is the emphasis on adolescents as typically leading sociolinguistic change. It is presumably important that it's these adolescents, in the Heritage Language context, who are experiencing the most intense contact. 

The term 'incomplete acquisition' is, of course, awkward and the authors are right to approach it with caution. However, it would be useful to interpret the phenomenon of the 'novel construction' in the light of 'incomplete acquisition', since something novel tends to replace something older, and it is the erosion of the older construction (still in normal use in non-heritage contexts) that gives rise to the faulty interpretation of incomplete acquisition. The findings provide an opportunity to deal with this terminological and conceptual issue.
